# Multiple Effects of Echinochrome A on Selected Ion Channels Implicated in Skin Physiology

**DOI:** 10.3390/md21020078

**Published:** 2023-01-23

**Authors:** Sung Eun Kim, Elina Da Sol Chung, Elena A. Vasileva, Natalia P. Mishchenko, Sergey A. Fedoreyev, Valentin A. Stonik, Hyoung Kyu Kim, Joo Hyun Nam, Sung Joon Kim

**Affiliations:** 1Department of Physiology, and Department of Biomedical Science, Seoul National University College of Medicine, Seoul 03080, Republic of Korea; 2Ischemic/Hypoxic Disease Institute, Seoul National University, Seoul 03080, Republic of Korea; 3G.B. Elyakov Pacific Institute of Bioorganic Chemistry, Far-Eastern Branch of the Russian Academy of Science, 690022 Vladivostok, Russia; 4Department of Physiology, College of Medicine, Cardiovascular and Metabolic Disease Center, Smart Marine Therapeutic Center, Department of Health Sciences and Technology, Graduate School, Inje University, Busan 47392, Republic of Korea; 5Department of Physiology, Dongguk University College of Medicine, Gyeongju 38066, Republic of Korea; 6Channelopathy Research Center (CRC), Dongguk University College of Medicine, Goyang-si 10326, Republic of Korea

**Keywords:** echinochrome A, skin, ion channel, TREK/TRAAK, TRPV3, orai1, TRPV1

## Abstract

Echinochrome A (Ech A), a naphthoquinoid pigment from sea urchins, is known to have anti-inflammatory and analgesic effects that have been suggested to be mediated by antioxidant activity and intracellular signaling modulation. In addition to these mechanisms, the ion channels in keratinocytes, immune cells, and nociceptive neurons may be the target for the pharmacological effects. Here, using the patch clamp technique, we investigated the effects of Ech A on the Ca^2+^-permeable TRPV3, TRPV1 and Orai1 channels and the two-pore domain K^+^ (K2P) channels (TREK/TRAAK, TASK-1, and TRESK) overexpressed in HEK 293 cells. Ech A inhibited both the TRPV3 and Orai1 currents, with IC_50_ levels of 2.1 and 2.4 μM, respectively. The capsaicin-activated TRPV1 current was slightly augmented by Ech A. Ech A alone did not change the amplitude of the TREK-2 current (I_TREK2_), but pretreatments with Ech A markedly facilitated I_TREK2_ activation by 2-APB, arachidonic acid (AA), and acidic extracellular pH (pH_e_). Similar facilitation effects of Ech A on TREK-1 and TRAAK were observed when they were stimulated with 2-APB and AA, respectively. On the contrary, Ech A did not affect the TRESK and TASK-1 currents. Interestingly, the I_TREK2_ maximally activated by the combined application of 2-APB and Ech A was not inhibited by norfluoxetine but was still completely inhibited by ruthenium red. The selective loss of sensitivity to norfluoxetine suggested an altered molecular conformation of TREK-2 by Ech A. We conclude that the Ech A-induced inhibition of the Ca^2+^-permeable cation channels and the facilitation of the TREK/TRAAK K2P channels may underlie the analgesic and anti-inflammatory effects of Ech A.

## 1. Introduction

Ion channels are pore-forming membrane proteins that are selectively or non-selectively permeable to various ions. Based on their physiological roles, ion channels have been intriguing targets for drug development. While their main mode of action is modulating the excitability of neurons and muscular organs, electrolyte transport mechanisms and Ca^2+^ influx-dependent signaling pathways in non-excitable epithelial and immune cells are also crucial targets of novel drugs, as well as various natural products [1,2,3].

The skin is the body’s largest organ, preventing excessive water loss and controlling the body temperature. In addition, it performs many vital functions, including protection against external physical, chemical, and biological assaults. The epidermis of the skin consists of multiple layers of keratinocytes, the functions of which are critical for protection and skin barrier formation [4]. The ion channels in keratinocytes, such as the transient receptor potential vanilloid (TRPV) family of cation channels and K^+^ channels, are drawing attention regarding their roles in differentiation, sensation, and inflammatory responses [5,6,7,8,9,10]. Among the TRPV family channels, it is well known that TRPV3 is involved in the differentiation of keratinocytes and its pathologic activity or genetic mutations are associated with skin disorders such as Olmsted syndrome [6,11,12,13,14,15,16,17]. In addition to TRPV3, the Ca^2+^-selective Orai1 channel is also known to play important roles in skin physiology and pathophysiology [18,19].

Ion channels in sensory nerve endings and certain types of immune cells in the skin also serve as interesting targets for controlling various skin diseases and peripheral pain [2,20,21,22,23]. A number of voltage-independent K2P K^+^ channel (KCNK) family with characteristic two-pore domains in the protomer, such as TREK, TRAAK, and TASK, are known to set the resting membrane potential of the nociceptive sensory neurons [24,25,26]. The K2P channels, noticeably TREK-2, are also expressed in skin keratinocytes [10] and immune cells [27,28]. However, the regulation of immune cell activation depends more on the Orai1/STIM1 complex than on other types of cation channels [29,30,31]. 

There has been a growing interest in identifying novel, effective, and safe molecules that can be used for cosmetics, as well as for the treatment of skin disorders. Natural products and their specific chemical substances are popular sources in the treatment of skin diseases. Their modes of action are anti-inflammatory, anti-bacterial, anti-pigmentating, and anti-wrinkle formation. The effects are mainly explained by biochemical signaling processes, including antioxidant activities [32,33].

Marine animal products such as echinochrome A (6-ethyl-2,3,5,7,8-pentahydroxy-1,4-naphthoquinone, Ech A) are potential sources of biologically active agents. Echinochrome A (6-ethyl-2,3,5,7,8-pentahydroxy-1,4-naphthoquinone, Ech A) is a pigment isolated from the internal skeleton of the sea urchin and is registered in the Russian pharmacopeia as an active drug substance (Histochrome^®^, G.B. Elyakov Pacific Institute of Bioorganic Chemistry FEB RAS, Vladivostok, Russia). It is used to treat inflammatory diseases of the eye, glaucoma, and cardiovascular diseases [34,35,36,37,38]. In addition, Ech A is known to have pharmacological effects on the experimental gastric ulcer model, and it changed the immune cell polarization in a colitis model animal [39,40]. Ech A is known to have beneficial effects against various diseases via its antioxidant activity and modulation of ROS-mediated signaling [34,35,36,37,38].

It has been reported that Ech A also affects skin-related symptoms such as fibrosis and inflammation in bleomycin-induced scleroderma [41]. In UVB-induced skin photoaging mice, Ech A injection lowered transepidermal water loss and attenuated tissue inflammatory changes and collagen degeneration [42]. In addition, Ech A alleviated atopic dermatitis (AD)-like skin lesions, inhibited mast cell infiltration, and reduced transepidermal water loss [43]. Unfortunately, the previously reported action mechanisms of Ech A in skin diseases have been limited to the biochemical signaling mechanisms. There has been no previous study on the pharmacological effects of Ech A on the ion channels expressed in the cells that compose skin tissue. 

Therefore, using the whole-cell patch clamp technique, we investigated the in vitro effects of Ech A on the ion channels that are known to be expressed in keratinocytes (TRPV3, Orai1, and TREK-2), immune cells (Orai1, TASK-1 and TREK-2), and nociceptive dorsal root ganglion (DRG) nerve endings (TREK/TRAAK, TRESK, TRPV1, and TRPV3). The results of our study firstly demonstrated the intriguing action of Ech A on the tested ion channels, which could suggest that it has a pharmacological mechanism in addition to its previously reported anti-inflammatory effects.

## 2. Results

### 2.1. Inhibitory Effects of Ech A on TRPV3 and Orai1

In a whole-cell configuration with a CsCl pipette solution, the TRPV3 current (I_TRPV3_) was induced by applying 2-aminoethoxydiphenylborate (2-APB, 100 μM), an agonist of TRPV3 [6]. A ramp-like depolarizing pulse of a voltage clamp (from −100 to 100 mV, 0.2 V/s) was applied every 20 s to obtain the current–voltage relation (I/V curve) of TRPV3 (Figure 1a,b). After confirming the steady-state maximum activity of TRPV3 with the linear shape of the I/V curve, 50 μM of 74a, a selective TRPV3 antagonist, was applied to confirm the complete inhibition of the current (Figure 1a,b). In other cells showing robust activation of I_TRPV3_, Ech A was added in increasing concentrations from 1 to 30 μM. The application of Ech A decreased the amplitude of I_TRPV3_ in a concentration-dependent manner (Figure 1c,d). At the end of each experiment, 50 μM of 74a was also applied to confirm the full inhibition of I_TRPV3_. A summary of the current density at −100 mV, as measured in the above experiments, is shown as bar graphs (Figure 1e). To obtain the profile of the inhibitory effects on I_TRPV3_, the normalized current density at −100 mV (I/Imax, %) fit to a logistic function is also shown (Figure 1f). The half-inhibitory concentration (IC_50_) of Ech A on I_TRPV3_ was 2.11 ± 1.055 μM (*n* = 6). Apart from 2-APB, I_TRPV3_ was induced by applying another agonist, drofenine (500 μM), and the current was also inhibited by 30 μM of Ech A (*n* = 5, data not shown).

To record the Ca^2+^-release-activated Ca^2+^ current, a CsCl pipette solution containing 20 μM of InsP_3_ and 20 mM of BAPTA was used for the whole-cell recording in HEK 293 cells overexpressed with Orai1 and STIM1 [29]. On making the whole-cell configuration, the spontaneous development of an inwardly rectifying current (I_Orai1_) was observed in the Orai1/STIM1 co-expressed HEK 293 cells (Figure 2a,b). After confirming a steady-state I_Orai1_, Ech A was applied in increasing concentrations from 1 to 30 μM. A concentration-dependent inhibition of I_Orai1_ was observed (Figure 2c,d). At the end of each experiment, 5 μM of GdCl_3_, a known inhibitor of I_Orai1_, was applied to confirm the full inhibition (Figure 2a,c). The summary of the current density at −130 mV as measured in the above experiments is shown as bar graphs (Figure 2e). The normalized current at −130 mV (I/I_max_) was obtained by subtracting the Gd^3+^-treated I/V curve from the maximum conductance (I_max_) after making the whole-cell configuration. To obtain the profile of the inhibitory effects on I_Orai1_, the normalized current density was fit to a logistic function. The IC_50_ of Ech A on I_Orai1_ was 2.41 ± 1.554 μM (Figure 2f, *n* = 6). 

### 2.2. Facilitating Effect of Ech A on TRPV1

A TRPV1 current (I_TRPV1_) was activated by applying capsaicin in the TRPV1 overexpressed HEK293 cells. At the holding voltage of −60 mV, the inward I_TRPV1_ was induced by 1 μM capsaicin. Interestingly, the I_TRPV1_ was increased, not decreased, by the additional application of 30 μM of Ech A (Figure 3a,b). The total inward current was abolished by a known TRPV1 inhibitor, BCTC (1 μM). Application of Ech A alone did not induce a significant effect on TRPV1.

### 2.3. Facilitation of TREK/TRAAK Activities by Ech A in the Presence of Agonists

To record the K^+^ channel current, a KCl pipette solution was used for the whole-cell recording in TREK-1 (KCNK2) or TREK-2 (KCNK10) overexpressing HEK 293 cells. Ramp-like depolarizing voltage clamp pulses were applied from −100 to 20 mV with −80 mV of holding voltage. I/V curves in the control condition showed an outward current with a hyperpolarized reversal potential of approximately −80 mV, indicating the activity of the K^+^ channels. The application of 30 μM of Ech A alone had no significant effect on the K^+^ channel currents. Apart from its agonist action on TRPV3, 2-APB is also known as a chemical agonist for TREK/TRAAK channels [44]. Consistently, 100 μM 2-APB increased the amplitude of both the TREK-1 (I_TREK1_) and TREK-2 (I_TREK2_) currents. Surprisingly, when combined with 30 μM of Ech A, the amplitudes of the currents were dramatically increased (Figure 4). The facilitation of 2-APB-induced I_TRKE2_ by Ech A was observed from the concentration of 3 μM. Unfortunately, however, a more rigorous pharmacological analysis of the concentration-dependent effect to obtain the half-effective concentration (EC_50_) could not be conducted since the facilitating effects at the relatively low concentrations were widely variable.

The TREK-1 and -2 and TRAAK channels are commonly activated by polyunsaturated fatty acids such as arachidonic acid (AA) [24,26]. The facilitating effect of Ech A was also observed when activated by arachidonic acid (AA, 10 μM). In TREK-2- or TRAAK overexpressed HEK 293 cells, the outward current activated by AA was markedly increased by the addition of 30 μM of Ech A (Figure 5). 

A unique feature of TREK-2 that was distinguishable from TREK-1 was its activation by acidic extracellular pH (pH_e_) [45]. Consistently, I_TREK2_ was increased when the pH_e_ was lowered from 7.4 to 5.5, and the application of Ech A further increased the outward current (Figure 6). However, the facilitating effect on the acidic pH_e_-induced I_TREK2_ was less significant than the responses of the 2-APB- or AA-induced I_TREK2_. 

### 2.4. Altered Sensitivity of TREK-2 to Norfluoxetine by Ech A 

Among the pharmacological agents acting on the TREK channels, a widely used anti-depressant fluoxetine and its active metabolite norfluoxetine (NFx) are known as effective inhibitors on both TREK-1 and -2 [46,47]. Although not the principal targets of the antidepressant, TREK channel inhibition by NFx has provided important insights into the conformational changes associated with the gating mechanism of the TREK channels [47,48]. Therefore, we examined whether the I_TREK2_ maximally activated by Ech A combined with 2-APB was still inhibited by NFx.

In the TREK-2-overexpressed HEK 293 cells, we confirmed that the I_TREK2_ activated by 100 μM 2-APB alone was almost completely inhibited by 10 μM of NFx (Figure 7a,b,e). In contrast, the large I_TREK2_ activated by the combined application of 2-APB and Ech A was resistant to NFx. Interestingly, an application of ruthenium red (RR, 50 μM) could completely abolish the I_TREK2_ by 2-APB and Ech A (Figure 7c,d,f).

### 2.5. No Changes in TASK-1 and TRESK Activity by Ech A 

Among the K2P family K^+^ channels, the TASK subfamily members are activated by an alkaline pH_e_. An RT-PCR analysis of HaCaT keratinocyte demonstrated the expression of TASK-1 and -3; therefore, we tested whether Ech A could also augment the TASK-1 (KCNK5) current. The outward K^+^ current in TASK-1-overexpressed HEK 293 cells (I_TASK1_) was increased by raising the pH_e_ from 7.4 to 8.5. However, different from the TREK/TRAAK channels, the amplitude of I_TASK1_ was not inhibited by Ech A (Figure 8a,b,e). 

Nociceptive DRG neurons express various types of K2P channels, among which, TREK-2 and TRESK (KCNK18) are major background K^+^ channels [25]. Thus, we also examined the effect of Ech A on the TRESK current (I_TRESK_). The TRESK-overexpressed HEK 293 cells showed a large outward K^+^ current in the control condition, and the application of up to 30 μM of Ech A had no significant effect on I_TRESK_ (Figure 8c,d,f).

## 3. Discussion

Our present study firstly showed multiple effects of Ech A on the representative ion channels expressed in keratinocytes, nociceptive neurons, and immune cells. Different from the inhibitory effects of the Orai1 and TRPV3 currents, we found a strong stimulated facilitation of the TREK/TRAAK subgroup of K2P channels by Ech A when applied together with the chemical activators. Further, the additional application of Ech A augmented the I_TRPV1_ inducted by the selective agonist capsaicin. 

### 3.1. Inhibitory Effect of Ech A on TRPV3 

TRPV3 is a thermosensitive nonselective cation channel primarily expressed in keratinocytes, oronasal squamous epithelia, and neurons [5,11,12]. TRPV3 activity is critical to the structure of, and for the function of, skin [13,14], and both loss- and gain-of-function mutations in human TRPV3 are associated with skin diseases such as Olmsted syndrome [15] and focal palmoplantar keratoderma [16]. We recently reported a novel high-frequency variant of TRPV3 p. A628T in East Asians which showed a faster sensitization in response to chemical agonists [17]. In addition, since TRPV3 is implicated in skin inflammation and irritative pruritic sensation [8], the investigation of TRPV3 inhibitors is drawing attention for the treatment of cutaneous sensory disorders [6,49,50]. In this respect, along with the plant-derived acridone alkaloid inhibitor of TRPV3 [49], the potent inhibition of TRPV3 by an animal-derived Ech A was an intriguing implication for the treatment of skin disorders and pruritis (Figure 1).

### 3.2. Inhibitory Effect of Ech A on Orai1

The inhibitory action of Ech A on the Ca^2+^-selective Orai1 channel in immune cells raised the additional pharmacological plausibility of relieving the symptoms of skin disorders. In various types of cells, the depletion of ER Ca^2+^ stores evoke a sustained Ca^2+^ influx called store-operated Ca^2+^ entry (SOCE). The electrophysiological characterization of SOCE revealed highly Ca^2+^- selective channels called Ca^2+^-release activated Ca^2+^ channels (CRAC). The CRAC current shows a distinctive inwardly rectifying current–voltage relation in mast cells and T cells [51]. As for the molecular identity of I_CRAC_, Orai1 is the Ca^2+^-conducting pore unit in the plasma membrane, and combined with STIM1 in ER membranes, it senses the Ca^2+^ filling state [30]. The Ca^2+^ influx via Orai1 is critical for the calcineurin activation and subsequent dephosphorylation of NFAT [30]. Therefore, the potent inhibitory effect of Ech A on the Orai1 activity (I_Orai1_, Figure 2) suggested that the anti-inflammatory action and therapeutic effects of Ech A on atopic dermatitis may be mediated, at least partly, by the inhibition of Ca^2+^-signaling in the immune cells. Although not an animal-derived molecule, curcumin has been reported to inhibit the CRAC current and SOCE in T lymphocytes [52]. A guava plant-derived molecule also has an anti-melanogenesis effect via Orai1 inhibition [53].

### 3.3. Strong Facilitation of TREK/TRAAK Channels by Ech A 

The KCNK channel family members share a characteristic topology of four transmembrane domains and two pore-forming loops. The homo- or hetero-dimeric compositions of KCNK channels form a functional K2P channel. K2P channels carry voltage-independent K^+^ currents in a variety of cell types, maintaining the resting membrane potential. The mammalian K2P family is made up of 15 members, which can be further divided into six subfamilies (TWIK, TASK, TREK/TRAAK, TRESK, THIK, and TALK) [24,26,54].

In contrast with the inhibitory actions on Ca^2+^-permeable TRPV3 and Orai1, Ech A induced the facilitation of TREK-1 (KCNK2) and -2 (KCNK10) and TRAAK (KCNK4) channels when combined with known agonists such as 2-APB and AA. 

Interestingly, the application of Ech A alone had no significant effect on the amplitude of the outward K^+^ current via TREK-1 and -2 and TRAAK. The channel activities of TASK-1 (KCNK3) and TRESK (KCNK18) were not affected by Ech A. Thus, the present results suggest that the facilitating effect of Ech A on the K2P channels appears to be limited to the members of the TREK/TRAAK subfamily.

It is well known that TREK-1 and TREK-2 activities can be increased by diverse physicochemical factors, such as AA, mechanical stretch, pH, PIP_2_, temperature, and pharmacological agents such as 2-APB. Such multi-modal agonistic sensitivity of the TREK channels allows them to integrate diverse cellular signals [26,55]. TREK channel mutations, especially their loss of function, are implicated in a range of pathological states, and vice versa, where their activation is involved in several protective mechanisms under ischemic damage and anesthesia [56]. TREK-1 knockout mice are more sensitive to painful heat and mechanical stimuli [57]. Furthermore, the siRNA knock-down of TREK-2 increased pain behaviors under skin inflammation [58]. TRAAK knockout mice also showed sensitivity to mechanical and heat stimuli [59] and postsurgical neuropathic pain [60].

Small-molecule TREK-1 activators have been reported to display antinociceptive properties in vivo [56]. These findings suggest that the TREK channels could be attractive targets for the development of novel therapeutics. Despite the recent development of activators and inhibitors targeting the TREK channels [60], further investigation of the potent and/or selective pharmacological tools for the TREK channels is awaited [26]. In this respect, although not a direct activator by itself, the strong facilitation by Ech A suggests a novel type of pharmacological tool for modulating the TREK channels.

### 3.4. State-Dependent Effects of Ech A on TREK-2

Among members of the TREK/TRAAK subfamily, we specifically focused on TREK-2 to analyze the pharmacological effects of Ech A since both keratinocytes and nociceptive DRG neurons express TREK-2 channels [10,25]. The common facilitatory effects of Ech A on the activation of TREK-2 by different activating conditions (2-APB, AA, and an acidic pH_e_) suggested that the mechanism may be associated with the open state of the channel. In other words, the binding site for Ech A may be accessible only when the conformation of TREK-2 is converted from a closed state to an open state of the channel. 

The principal gating mechanism of TREK is the “C-type” gate, a process by which ion flux through a K^+^ channel is regulated at the selectivity filter [47]. Molecular structural studies combined with electrophysiological analysis revealed that the fourth transmembrane (M4) helix of TREK is mobile and can adopt conformations that range between an “up” state and a “down” state of the channels. Although direct evidence is lacking, the agonist-dependent different effects of Ech A suggest that its binding with TREK-2 may occur at the upstate of the channel (see below).

Recent molecular structural studies on TREK channels have also revealed intriguing pharmacological features related to different sites of the channel molecule for binding with small molecules. From the outside to the inside direction of TREK, (1) the Keystone inhibitor site [61], (2) the K2P modulator pocket [62], (3) the fenestration site [63,64], and (4) the modulatory lipid site [47,62] have been reported. At the outermost extracellular domain of the TREK channel, a unique structure called the CAP domain can be found. It is an arch directly over the channel pore which creates the bifurcated extracellular ion pathway (EIP) from which the ions exit the channel after passing through the selectivity filter [47].

Regarding the polysite pharmacological property of TREK-2, the altered sensitivity of TREK-2 to the chemical inhibitor NFx is an intriguing result in our present study. While the TREK-2 current activated by 2-APB or AA alone was almost completely inhibited by NFx, the current facilitated by Ech A became less sensitive to NFx, though it was totally inhibited by RuR (Figure 7). RuR is a trinuclear oxo-bridged ruthenium amine with polycationic charges. RuR interacts with the negatively charged residues in the Keystone inhibitor site at the base of the CAP domain. In contrast, NFx binds with the fenestration site. In this respect, the altered sensitivity to NFx suggests a possible Ech A-mediated conformational change in TREK-2 that is less accessible to NFx. As mentioned above, TREK-2 has “up” and “down” positions of the M4 helix. The “down” state creates the fenestration site just below the second pore helix (P2) that is open to the center of the membrane [48]. The TREK inhibitor NFx binds with the fenestration site defined by the lower part of the P2 pore helix and the “down” state M4 [63]. Based on the known structural properties of the TREK channels, the lowered sensitivity to NFx implies a putative shift from the “down” to the “up” state of TREK-2, which could be associated with Ech A. 

The possibility that Ech A may alter the properties of the plasma membrane itself is a further consideration. A recent study indicated that phytochemicals with a phenol structure can penetrate the phospholipid bilayer and alter its structure, thereby affecting the function of membrane proteins [65]. Likewise, Ech A may act on the TREK channels because the channels can be activated by structural asymmetry induced by membrane stretch upon an application of physical pressure or arachidonic acid (AA). However, Ech A alone cannot elicit any response from the TREK channels, and its potentiation effect occurs in the presence of an agonist such as AA and 2-APB. It is believed that a complex mechanism beyond simple a structural change is involved in the pharmacological effect of Ech A on TREK-1 and TREK-2.

### 3.5. Facilitation of TRPV1 by Ech A

Along with the facilitation of the TREK/TRAAK family K^+^ channels, the augmentation of I_TRPV1_ by Ech A has drawn attention. Because TRPV1 is the representative nociceptive channels [66], its facilitation may imply a putative irritative effect by Ech A. However, previous studies have not yet reported such a symptom or irritative side effect. 

Ech A is a quinoid compound that can be quite reactive to various biological molecules. Although we have not directly examined the skin toxicity of Ech A, a large number of studies, including clinical applications such as a topical injection of Histochrome^®^, did not report noticeable toxicity [34,35,36,37,38]. A recent study by Mishchenko et al. suggested that the toxicity of Ech A and its metabolites are insignificant [67].

## 4. Materials and Methods

### 4.1. Cell Culture

The human embryonic kidney cell line HEK 293 (CRL-3216, ATCC, Manassas, VA, USA) was used for the electrophysiological recordings. The HEK 293 cells were maintained in DMEM supplemented with 10% FBS and 1% penicillin-streptomycin. The HEK 293 cells were transiently transfected with the TRPV1, TRPV3, Orai1/STIM1, and K2P channels using a Turbofect transfection reagent (ThermoFisher Scientific, Waltham, MA, USA). For the Orai1 current recording, HEK293 cells were co-transfected with the human Orai1 (hORAI1) and human STIM1 (hSTIM1) vectors that were purchased from ORIGENE Technologies (Rockville, MD, USA). The complementary DNAs of the rTREK-1 (rKcnk2, NM172042), rTREK-2 (rKcnk10, NM023096), and hTRAAK (hKCNK4, NM033310) were purchased from ORIGENE (Rockville, MD, USA). The human pcDNA-TASK-1 (NM002246) plasmid was generously given to us by Dr. P. R. Stanfield (University of Leicester, Leicester, UK). The human TRPV3 (pReceiver -M02) was purchased from Genecopoeia (Rockville, MD, USA). The human TRPV1 (hTRPV1, pcDNA5/FRT) was provided by Professor Joo Hyun Nam (Dongguk University, Gyeongju, Korea). The plasmid DNA encoding mouse TRESK (mKCNK18, pcDNA 3.1) was provided by Professor Da Won Kang (Gyeongsang National University, Jinju, Korea). 

### 4.2. Chemicals

Histochrome^®^ was generated as described earlier [67]. The composition of Histochrome^®^ contains only Ech A with sodium carbonate (in ratio of 1:0.4) in a water solution (10 mg/mL). The preparation is known to be stable in sealed ampoules for several years [67]. After opening the ampoule, the Histochrome^®^ was kept in a tightly sealed condition with light protection and used as a stock solution to perform the experiments. Immediately before the experiments, the stock solution was taken out of the freezer to prepare the EchA-containing bath perfusing solution. The Histochrome^®^ was diluted in a bath solution in a concentration range of less than 0.1% for the purposes of the experiments. All the other chemicals were purchased from Sigma-Aldrich (St. Louis, MO, USA).

### 4.3. Electrophysiological Recording

The cells were transferred to a perfusion chamber mounted on the stage of an inverted microscope (Nikon TE2000-S, Tokyo, Japan). The whole-cell patch-clamp recordings were performed using a patch-clamp amplifier (Axopatch-200B; Axon Instruments, Foster City, CA, USA). We used a Digidata-1322A (Axon Instruments) interface and pCLAMP software (v.10.3) to apply the command pulses and acquire the data. The current recordings were made at room temperature (23–25 °C) using glass microelectrodes with a resistance of approximately 1.2–1.8 MΩ. The microglass pipettes (World Precision Instruments, Sarasota, FL, USA) were fabricated using an MF-830 microforge (Narishige, Tokyo, Japan). The data were low-pass-filtered at 2 kHz and digitally sampled at 10 kHz. The intracellular pipette solution for the TRPV3 current recording contained (in mM): 140 CsCl, 10 HEPES, 10 EGTA, 4.85 CaCl_2_ and 3 MgATP (adjusted to a pH of 7.2 with CsOH). The extracellular bath solution contained for TRPV3 current recording contained (in mM): 139 NaCl, 5 KCl, 10 HEPES, 3 BaCl_2_, 10 glucose, 2 MgCl_2_ and 1 EGTA (adjusted to pH 7.4 with NaOH). For TRPV1 current recording, the intracellular solution contained (in mM): 140 CsCl, 10 HEPES, 1 EGTA, 4 NaCl, 5 MgATP (adjusted to pH 7.2 with CsOH). The extracellular bath solution for the TRPV1 current recording contained (in mM): 140 NaCl, 4 KCl, 10 HEPES, 20 sucrose, 5 glucose, 1 EGTA and 1 MgCl_2_ (adjusted to pH 7.4 with NaOH). The intracellular pipette solution for the Orai1 current recording contained (in mM): 130 Cs-glutamate, 20 HEPES, 0.002 sodium pyruvate, 20 BAPTA, 0.02 InsP_3_, 1 MgCl_2,_ and 3 MgATP (adjusted to a pH of 7.2 with CsOH). The extracellular bath solution contained (in mM) 135 NaCl, 3.6 KCl, 10 HEPES, 5 glucose, 1 MgCl_2_ and 10 CaCl_2_ (adjusted to a pH of 7.4 with NaOH). The internal solution used for the K2P current recording contained (in mM): 6 NaCl, 135 KCl, 10 HEPES, 5 EGTA, and 3 MgATP (adjusted to a pH of 7.2 with KOH). The bath solution contained (in mM): 145 NaCl, 3.6 KCl, 1.3 CaCl_2_, 1 MgCl_2_, 5 glucose, and 10 HEPES (adjusted to a pH of 7.4 with NaOH). The bath solution was then perfused at 2 mL/min at room temperature. After the current was recorded stably and the response was shown, the bath solution was replaced immediately. 

### 4.4. Statistical Analysis

All data are expressed as mean ± standard errors of the mean, and the statistical analysis was performed by one-way analysis of variance (ANOVA) and Tukey’s post-hoc tests. Statistical significance was defined as * *p* < 0.05, ** *p* < 0.01, *** *p* < 0.001, and **** *p* < 0.0001. Origin pro 2016 and GraphPad Prism 8.0.1 were used for the statistical analysis and data visualization.

## 5. Conclusions

We are the first researchers to report the pharmacological effects of Ech A on ion channels. We showed an inhibitory action of Ech A on TRPV3 and Orai1, while Ech A facilitated the activation of TREK/TRAAK when stimulated by their chemical agonists. Such an intriguing action of Ech A on ion channels suggests a pharmacological mechanism for skin diseases in addition to its previously reported anti-inflammatory effects. Considering the recently reported therapeutic effects of Ech A on various skin diseases [41,42,43] and nociception [68], it would be an interesting theme to further investigate the pharmacological mechanism mediated by the action of Ech A on ion channels and Ca^2+^ signaling.

## Figures and Tables

**Figure 1 marinedrugs-21-00078-f001:**
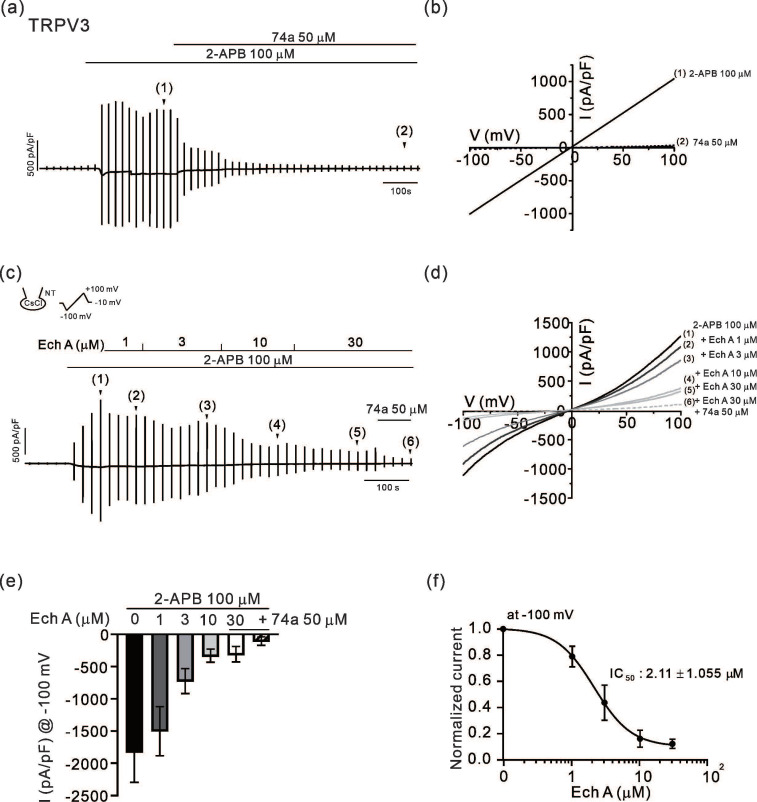
Effects of Ech A on I_TRPV3_. (**a**,**c**) An original trace of the membrane current density (pA/pF) demonstrating the activation of TRPV3 by 100 μM 2-APB and its inhibition by 74a (**a**) or by Ech A (**c**). The vertical lines indicate the current density obtained by ramp-like pulses from −100 to 100 mV (holding voltage, −10 mV) applied with an interval of 20 s. The I/V curves are recorded at the points indicated by the numbers in parentheses (**b**,**d**). (**e**) A summary of the inward current densities (pA/pF at −100 mV) activated by 2-APB and with Ech A or 74a (*n* = 6). (**f**) A semi-logarithmic plot of the normalized current according to the incremental concentrations of Ech A, fit with a logistic function (IC_50_, 2.11 ± 1.055 μM, *n* = 6). The inward current at −100 mV was normalized to the maximum I_TRPV3_ (the peak inward current subtracted with the residual inward current in the presence of 74a) in each cell.

**Figure 2 marinedrugs-21-00078-f002:**
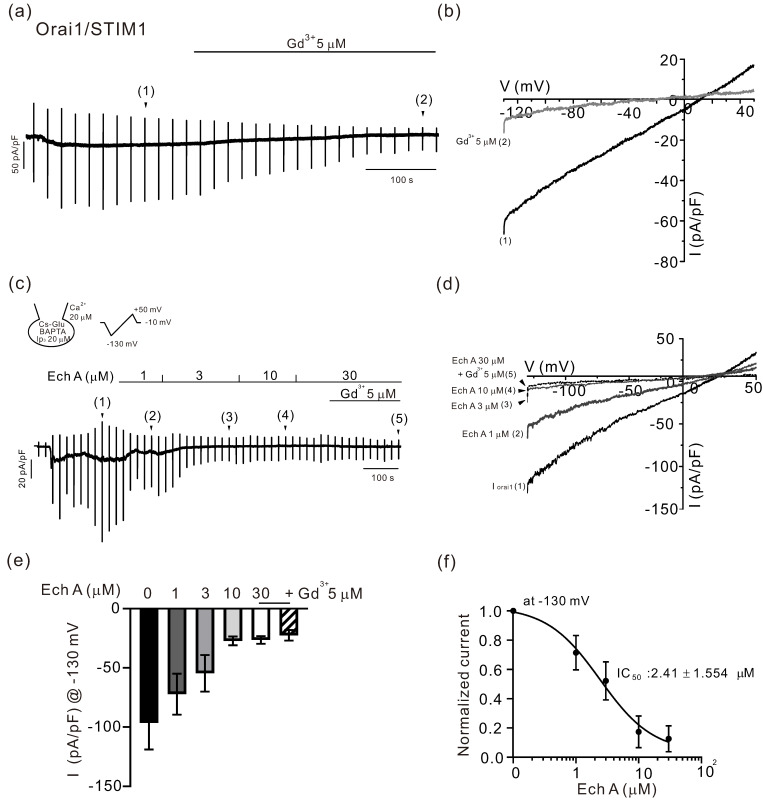
Inhibitory effects of Ech A on I_Orai1_. (**a**,**c**) Representative traces of membrane current density (pA/pF) with responses to the repetitive ramp-like voltage clamp pulses from −130 to 50 mV, 0.2 V/s (vertical lines), with a CsCl pipette solution containing 20 μM of InsP_3_ and 20 mM of BAPTA. After making a whole-cell configuration, the spontaneous activation of I_Orai1_ was observed, and the concentration-dependent inhibition of I_Orai1_ by Ech A (**c**) or by 5 μM of GdCl_3_ (**a**,**c**) was confirmed. (**b**,**d**) The I/V curves recorded at the points are indicated by the numbers in parentheses. (**e**) A summary of the inward current densities (pA/pF) measured at −130 mV in the control and with the incremental concentrations of Ech A (*n* = 6). (**f**) A semi-logarithmic plot of the normalized current according to the concentrations of Ech A and fit with a logistic function (IC_50_, 2.41 ± 1.554 μM, *n* = 6). The inward current at −130 mV was normalized to the maximum I_Orai1_ (the peak inward current subtracted with the residual inward current with GdCl_3_) in each cell.

**Figure 3 marinedrugs-21-00078-f003:**
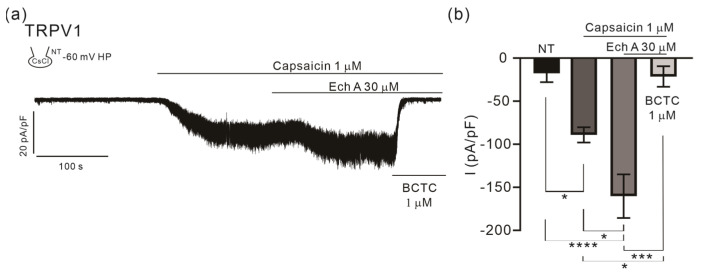
Effects of Ech A on I_TRPV1_. (**a**) An original trace of the membrane current density (pA/pF) recorded at the holding voltage of −60 mV, demonstrating the activation by capsaicin (1 μM) and the additional increase by the combined application of Ech A (30 μM). The total inhibition by BCTC (1 μM) was confirmed at the end of experiment. (**b**) A summary of the inward current density of the control (NT), I_TRPV1_ activated by capsaicin, further increased by Ech A, and inhibition by BCTC (*n* = 5). Data represent the mean ± SEM. * *p* < 0.05; *** *p* < 0.001; **** *p* < 0.0001.

**Figure 4 marinedrugs-21-00078-f004:**
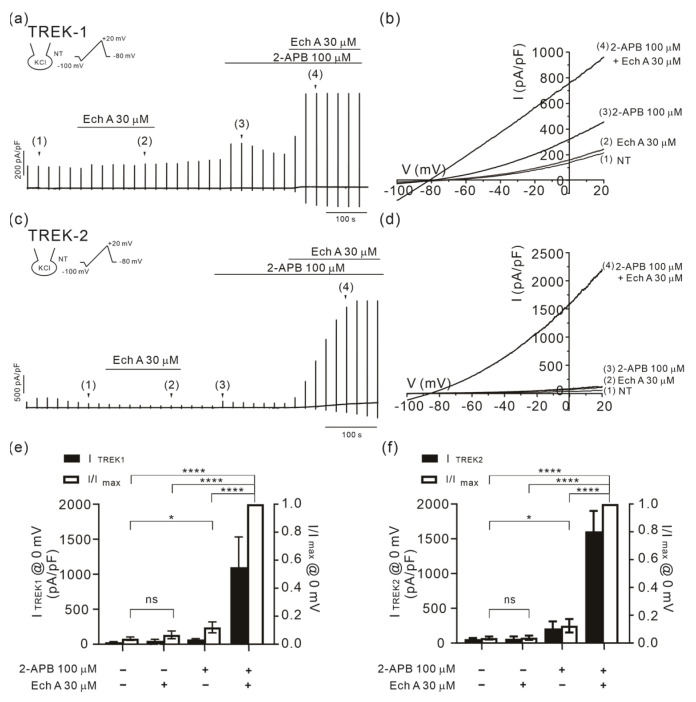
Facilitating effects of Ech A on I_TREK1_ and I_TREK2_ with 2-APB. (**a**,**c**) Representative traces of the membrane current density (pA/pF) recorded with a KCl pipette solution in TREK-1- (**a**) and TREK-2- (**c**) overexpressed HEK 293 cells. I/V curves were obtained by ramp-like depolarizing pulses from−100 to 20 mV (**b**,**d**). (**e**,**f**) A summary of the outward current densities (pA/pF, left vertical axis) measured at 0 mV in the control, 2-APB (100 μM), Ech A (30 μM), and 2-APB+ Ech A (closed bars). The normalized current (I/Imax at 0 mV, right axis) against the peak amplitude induced by the combined application of 2-APB and Ech A is also shown (open bars). *n* = 6 for TREK-1 (**e**) and *n* = 7 for TREK-2 (**f**). Data represent the mean ± SEM. * *p* < 0.05; **** *p* < 0.0001; ns, statistically not significant.

**Figure 5 marinedrugs-21-00078-f005:**
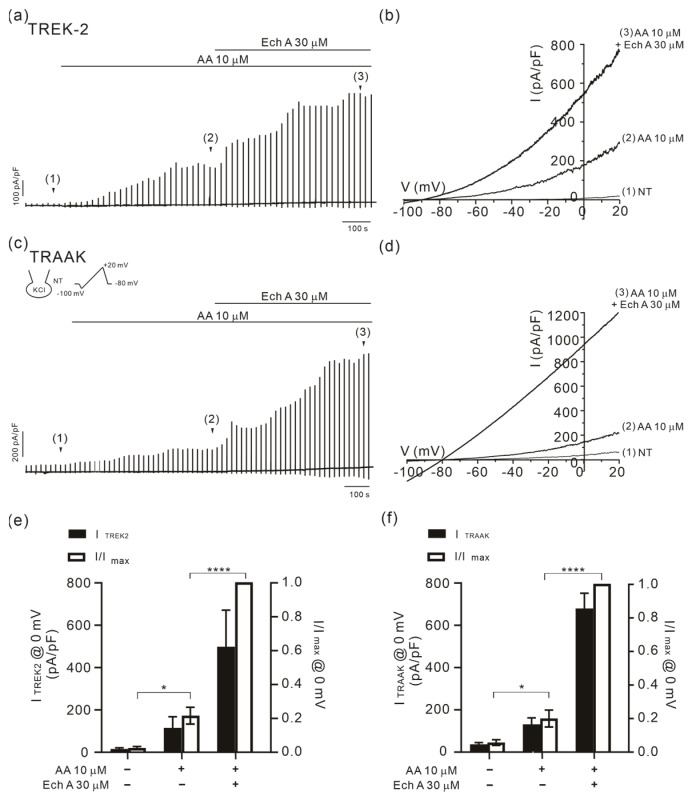
Facilitating effects of Ech A on I_TREK2_ and I_TRAAK_ with AA. (**a**,**c**) A representative traces of the membrane current density (pA/pF) in TREK-2- (**a**) and TRAAK- (**c**) overexpressed HEK 293 cells. I/V curves were obtained by ramp-like depolarizing pulses from −100 to 20 mV (**b**,**d**). (**e**,**f**) A summary of the outward current densities (pA/pF, left vertical axis) measured at 0 mV in the control, AA (10 μM), Ech A (30 μM), and AA + Ech A (closed bars). The normalized current (I/Imax at 0 mV, right axis) against the peak amplitude induced by the combined application of 2-APB and Ech A is also shown (open bars). *n* = 5 for TREK-2 (**e**) and *n* = 5 for TRAAK (**f**). Data represent the mean ± SEM. * *p* < 0.05; **** *p* < 0.0001.

**Figure 6 marinedrugs-21-00078-f006:**
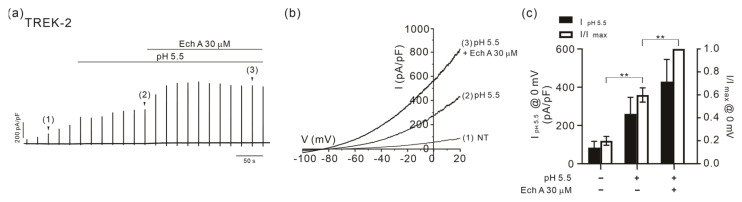
Facilitating effects of Ech A on I_TREK2_ activated by acidic pH_e_. (**a**) A representative trace of the membrane current in TREK-2-overexpressed HEK 293 cells. (**a**) I/V curves were obtained by ramp-like depolarizing pulses from −100 to 20 mV (**b**). (**c**) A summary of the outward current densities (pA/pF, left vertical axis) measured at 0 mV in the control, pH_e_ 5.5, and Ech A (30 μM) combined with pH_e_ 5.5 (closed bars). The normalized current (I/Imax at 0 mV, right axis) against the peak amplitude induced by the combined application of pH_e_ 5.5 and Ech A is also shown (open bars, *n* = 4). Data represent the mean ± SEM. ** *p* < 0.01.

**Figure 7 marinedrugs-21-00078-f007:**
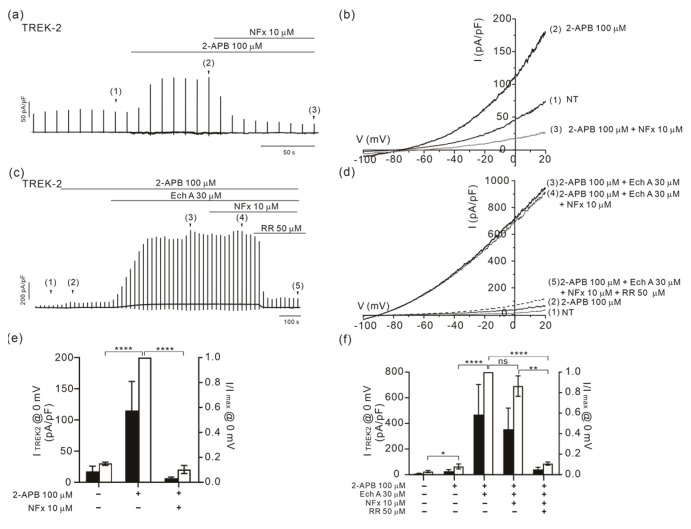
Disappearance of the inhibitory effect of NFx on the I_TREK2_ facilitated by Ech A. (**a**,**b**) A representative current trace and I/V curve of I_TREK2_ activated by 100 μM 2-APB alone, which was almost completely inhibited by 10 μM of NFx. (**c**,**d**) A representative current trace and I/V curve of I_TREK2_ activated by 2-APB and 30 μM of Ech A, which was not inhibited by 10 μM of NFx. The I/V curves were obtained by ramp-like depolarizing pulses from −100 to 20 mV. (**e**,**f**) A summary of the outward current densities (pA/pF, left vertical axis) measured at 0 mV in the control, 2-APB, and NFx (closed bars). A summary of the normalized current (I/Imax at 0 mV, right axis) against the peak amplitude is also shown (open bars). The inhibition by 50 μM RR is also shown in the panel (**f**). *n* = 5 for (**e**) and *n* = 4 for (**f**). Data represent the mean ± SEM. ** *p* < 0.01; **** *p* < 0.0001; ns, not statistically significant.

**Figure 8 marinedrugs-21-00078-f008:**
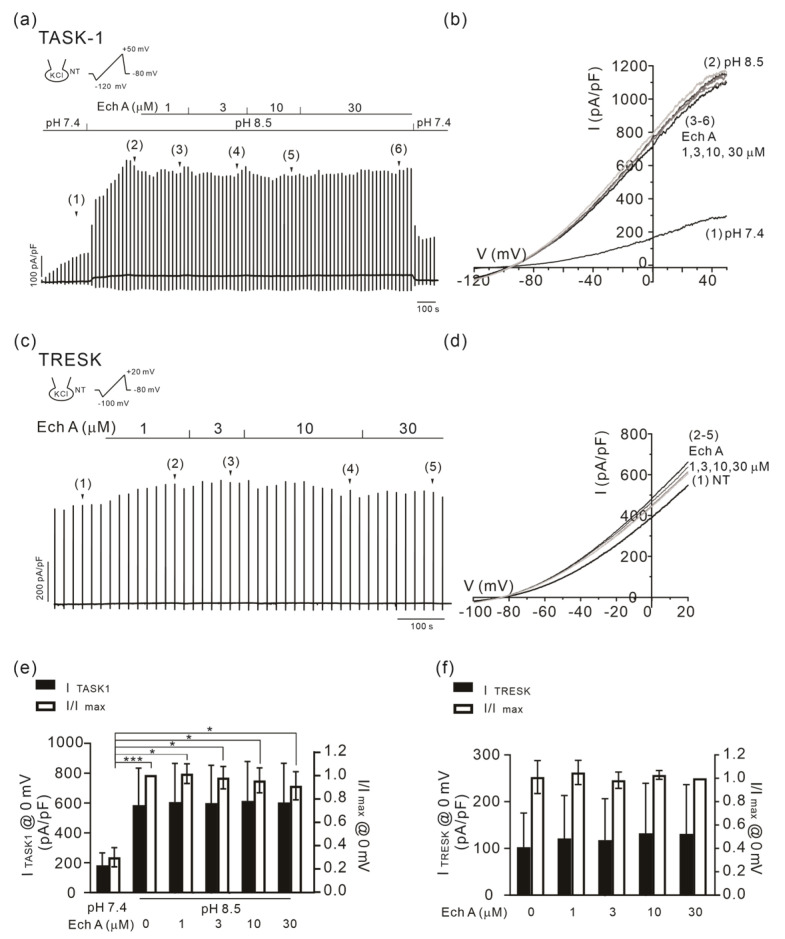
Effects of Ech A on I_TASK1_ and I_TRESK_. (**a**,**b**) A representative current trace and I/V curve of I_TASK1_ activated by an alkaline pHe of 8.5 and the partial inhibitory effect of Ech A. (**c**,**d**) A representative current trace and I/V curve of I_TRESK_. The application of up to 30 μM of Ech A had no significant effect. The I/V curves were obtained by ramp-like depolarizing pulses from −120 to 50 mV (vertical lines in (**a**)) and from −100 to 20 mV (vertical lines in (**c**)). (**e**,**f**) A summary of the outward current densities (pA/pF, left vertical axis) measured at 0 mV (closed bars). A summary of the normalized current (I/Imax at 0 mV, right axis) against the peak amplitude is also shown (open bars). *n* = 9 for TASK-1 (**e**) and *n* = 4 for TRESK (**f**). Data represent the mean ± SEM. * *p* < 0.05; *** *p* < 0.001.

## Data Availability

The data presented in this study are available on request from the corresponding author.

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
