# Peer review of "Multiple Effects of Echinochrome A on Selected Ion Channels Implicated in Skin Physiology"

_marinedrugs, 2023, doi:10.3390/md21020078_

Round 1

Reviewer 1 Report

I have read the manuscript and I have some questions and recommendations.

1. The title of the manuscript should correspond to the purpose and conclusions. Give the title of the manuscript. What is "Multimodal Effects"? This term occurs once in the text (line 293).

2. Echinochrome A, a pigment originally isolated from the internal skeleton of a sea urchin. To increase interest in echinochrome, please indicate in the introduction that echinochrome is registered as a medicinal product [https://doi.org/10.1016/j.jep.2019.111933], the effectiveness of which has been studied and proven.

3. In Section 4 "Materials and Methods" it is necessary to indicate the source of echinochrome A, its purity, series.

4. Please specify the solvent/carrier for the experiment.

5. No positive control. A positive control group should be included.

6. There are no statistical differences in Figure 7. Provide data.

7. In section 4 "Materials and methods" it is necessary to give information about the methods, assortment. Give references or give validation characteristics.

8. The quality of the figures is unsatisfactory.

9. The pharmacokinetics of natural molecules is extremely complex. The pharmacokinetics of marine drugs has attracted increasing interest in recent decades due to its effective and potential contribution to the selection of rational dosage recommendations and the optimal use of the therapeutic arsenal. From the point of view of the future use of echinochrome, it is very important that pharmacokinetic data are available for it (for example, https://doi.org/10.3390/md18110557). The pharmacokinetics of echinochrome when applied to the skin has been studied?

10. What is the toxicity of Echinnochrome A when applied topically? Provide data or literary reference.

11. Indicate the number of the protocol of the bioethical commission that approved this study.

Author Response

We would like to thank the editor and the reviewers for their constructive comments and critical reviews. We conducted the additional experiments suggested from the reviewers and have made the required changes, as detailed below.

Reviewer #1

  1. The title of the manuscript should correspond to the purpose and conclusions. Give the title of the manuscript. What is "Multimodal Effects"? This term occurs once in the text (line 293).

Response: Thank you for this comment. We have changed the title of our manuscript to: 
“Multiple effects of Echinochrome A on the ion channels expressed in skin tissues” We hope that this title is now suitable as per the concern of the reviewer.

  1. Echinochrome A, a pigment originally isolated from the internal skeleton of a sea urchin. To increase interest in echinochrome, please indicate in the introduction that echinochrome is registered as a medicinal product [https://doi.org/10.1016/j.jep.2019.111933], the effectiveness of which has been studied and proven.

Response: Thank you for this suggestion. We have changed the introduction as follows, and have also added the reference recommended by you (p.2).
“Echinochrome A (6-ethyl-2,3,5,7,8-pentahydroxy-1,4-naphthoquinone, Ech A) is a pigment isolated from the internal skeleton of sea urchin and registered in the Russian pharmacopeia as an active drug substance (Histochrome®). It is used to treat cardiovascular diseases, glaucoma, and inflammatory diseases of the eye in Russia. Ech A is also known to have beneficial effects against various diseases via its antioxidant activity and modulation of ROS-mediated signaling [36-38].

  1. In Section 4 "Materials and Methods" it is necessary to indicate the source of echinochrome A, its purity, series.
    4. Please specify the solvent/carrier for the experiment.

Response to the comment No.3 and 4: Thank you for pointing this out. When we initially submitted the manuscript, we did not include a large part of the content in the Materials and Methods section. We apologize for this oversight. The Materials and Methods section has been revised thoroughly, and the original source, and information regarding the stock solutions for echinochrome A have been added (p.11).

  1. No positive control. A positive control group should be included.

Response: According to the comment, we added positive control data on the TRPV3 and Orai1 currents with the inhibitory effects by 74a and Gd3+, respectively (revised Fig. 1(a), (b) and Fig. 2(a), (b)). The corresponding description was included in the revised Results section (p.3 and 4). It has to be admitted that the stability of Orai1 current was not perfectly consistent. Therefore, one has to be careful about the interpretation of the pharmacological analysis of Ech A on the IOrai1, which is also briefly mentioned in the revised Results section (p. 4).

  1. There are no statistical differences in Figure 7. Provide data.

Response: We conducted additional experiments to clarify the statistical significance, and could conclude that Ech A had no significant effect on the TASK-1 and TRESK currents (revised Fig. 7). Although our previous data shown in the original manuscript suggested a tendency of inhibitory effect of Ech A on the TASK-1 current, the tendency was not significant after increasing the numbers of data. Therefore, the corresponding Results section was changed reflecting the experiments. I greatly appreciate the critical comment by the reviewer.

  1. In section 4 "Materials and methods" it is necessary to give information about the methods, assortment. Give references or give validation characteristics.

Response: As mentioned in our response to the point 4, we have added the missing information to the Materials and Methods section now. We again apologize for this issue.

  1. The quality of the figures is unsatisfactory.

Response: According to the comment, we have added positive control data (Fig. 1 and 2) and conducted additional experiments to obtain the better quality of figures. The representative figures were also changed in Fig. 6 and Fig. 7 in the revised manuscript.

  1. The pharmacokinetics of natural molecules is extremely complex. The pharmacokinetics of marine drugs has attracted increasing interest in recent decades due to its effective and potential contribution to the selection of rational dosage recommendations and the optimal use of the therapeutic arsenal. From the point of view of the future use of echinochrome, it is very important that pharmacokinetic data are available for it (for example, https://doi.org/10.3390/md18110557). The pharmacokinetics of echinochrome when applied to the skin has been studied?

Response:
Thank you for this comment and query. EchA is used clinically in Russia, and pharmacokinetic studies have been conducted with intravenous and oral injections. However, pharmacokinetic data for topical and subcutaneous application of EchA are currently unavailable. However, the chemistry and pharmacokinetics of EchA after oral administration revealed that it complied with the Lipinski Rule of Five (https://www.ncbi.nlm.nih.gov/pmc/articles/PMC8151293/pdf/marinedrugs-19-00267.pdf). Therefore, topical and subcutaneous pharmacokinetics may also satisfy the conditions for drug use. However, further research is warranted on this.

  1. What is the toxicity of Echinnochrome A when applied topically? Provide data or literary reference.

Response: Thank you for this relevant comment. There is one study related to toxicity of Ech A in the skin. Although the toxicity study was not conducted with a single component of EchA, it was confirmed that there was no itching effect when a local skin irritation test was performed in rabbits using the sea urchin pigment extract (PMID:24288292 DOI:10.1055/s-0033-1351098).  Moreover, in silico toxicity analysis of, Ech A and its oxidative degradation products, using the ProTox-II webserver revealed that EchA and its derivatives appear to have lower toxicity (≥2000 mg/kg) (https://www.ncbi.nlm.nih.gov/pmc/articles/PMC7587531/pdf/molecules-25-04778.pdf ). However, more vigorous investigations need to be conducted in future in in vivo models.

  1. Indicate the number of the protocol of the bioethical commission that approved this study

Thank you for your comments. We have added this information in the revised the Materials and Methods section, including the details of the bioethical commission that approved this study.

Author Response

Reviewer 2

The manuscript by Kim and colleagues describes a survey of several ion channels for their sensitivity to a polyhydroxylated 1,4-naphthoquinone enriched in sea urchins, which they called Ech A. They used the HEK cells model system to express each channel cDNA, patch-clamp recorded current responses to activation stimuli while adding varying amount of Ech A. They observed an inhibitory effect of Ech A on TRPV3 and Orai1/STEM channels at low micromolar concentrations, a potentiating effect on TREK-1 and TREK-2 at 30 micromoles, and minor effects on TASK-1 and TRESK at up to 30 micromoles concentrations. Previous studies have indicated that Ech A exerts antioxidant activities and modulates ROS-mediated signaling. The present study expands the list of potential targets for Ech A, which has attracted interests in applications for cosmetics and skin disease treatments.

Here is a list of issues the authors should address to improve the study:

  1. Motivation of the study. It is stated that ion channels related to skin physiology were selected for this study. However, the group of ion channels do not form a complete list for the goal, and some are marginally related to skin health. Meanwhile, TRPM4 has been found to be involved in skin health, with human mutations leading to a severe disease (DOI: 10.1016/j.jid.2018.10.044; 10.3389/fimmu.2022.1025499), but it is not tested. Rationale for the study needs be improved.

Response: Thank you for this critical observation. As the reviewer stated, various ion channels, including the TRP family channel, are expressed in the skin cells, and are known to play various physiological and pathophysiological roles. Due to the limit of time and scope of the study, here we focused to a limited number of ion channels that are expected to have a direct effect on skin inflammation. In particular, in the case of TRPM4, as suggested by the reviewer, we are aware that gain-of-function mutations in humans can cause progressive symmetric erythrokeratodermia owing to its hyperactivity. However, according to our literature search, the relevance between TRPM4 and skin disease has not be widely reported yet. Nevertheless, we agree that further in-depth research is required in future, to analyze the regulatory action of EchA on various ion channels, including TRPM4.

  1. Methods. Being a lipophilic molecule, Ech A is expected to have very low water solubility. Information on the type of solvent used, and its final concentration in working solutions, should be provided in the Methods section.

Response: Thank you for this comment. When we initially submitted the manuscript, we did not include a large part of the content in the Materials and Methods section. We apologize for this oversight on our part. The Materials and Methods section has been revised thoroughly, and the original source, and information regarding the stock solutions for echinochrome A have been added.

  1. Experimental design. When testing Ech A on TRPV3 and Orai1 currents, increasing concentrations were used while the current was declining (Figures 1&2). For this situation, potential channel desensitization should be examined and its extent (if observed) should be reported. Control recordings for the same duration upon 2-APB stimulation should be carried out, and/or randomization of the application sequence of different Ech A concentrations should be used.

Response: Thank you for this very useful comment. We have added positive control data to the ORAI1 and TRPV3 overexpressed HEK-293T cells. The ORAI1 current activated by ER depletion was confirmed to be well maintained and effectively inhibited by EchA. It was also confirmed that the TRPV3 current was well maintained without inactivation and effectively inhibited by EchA. In addition, specific agonist-induced ionic currents to each ion channel (ORAI1, TRPV3, and TREKs) were observed to be effectively blocked by their specific blockers.

  1. Experimental design. To confirm the potentiation effects of Ech A on TREK-1 and TREK-2, it is important to rule out the possibility of developing a gradual leaky patch. Had the channel inhibitor RR (Figure 6c) been used for both channels?

Response: Thank you very much for this query. When a gradual leaky patch is formed while the K2P current is activated, the reversal potential should gradually depolarize on the I-V curve because of the increase in the nonselective ionic current. However, in our experiment, it was observed that a reversal potential was formed close to the equilibrium potential of K+ ions in the current generated by the activation of the K2P channel. Therefore, the possibility of a gradual leaky patch is quite low. As you have mentioned the same experiment was performed on TREK-1, and the relevant data are presented in the revised figure. And, yes, we confirmed TREK-1 and TREK-2 inhibition using RR.

  1. Data interpretation. Given that the ion channels exhibiting sensitivities to Ech A have distinct structures and gating mechanisms, it is possible that Ech A, again being strongly lipophilic, may modulate channel activities through a membrane effect, especially when used at 30 micromoles (see DOI: 10.1021/cb500086e). This possibility should be considered and discussed.

Response: Thank you for your comments. We carefully read the paper suggested by the reviewer. Also worth considering is the possibility that the phytochemical itself can alter the membrane structure. We added the possibility with relevant reference at the end of the revised Discussion (p. 12).

Reviewer 3 Report

The paper by Kim et al describes the modulatory effect of a marine compound Echinochrome A (Ech A) from sea urchins on a panel of ion channels expressed in HEK293 cells using whole-cell patch clamp recordings.  Authors seemed to hypothesize that the multifunctional compound previously known and used to treat skin inflammation and disorders may also affect ion channels robustly expressed in the skin. They found that EchA inhibits the heat-sensitive Ca2+ permeable TRPV3 and Orail currents with IC50s in the range about 2.0 mM, in which Ech A seems to be a relatively potent and an interesting natural inhibitor. Ech A also facilitates the activation of TREK-1 and TRAAK K2P channels. Overall, authors presented some interesting and novel data. Here are some specific comments/suggestions for this paper:

1. The inhibition of TRPV3 by Ech A looked quite slow. How specific for Ech A affecting or inhibiting some other members of thermoTRPV subfamily such as TRPV1, 2 and 4? It will be interesting to know if Ech A can be used as a tool molecule for the TRPV subfamily.

2. Related to above, can Ech A inhibit TRPV3 current activated by other specific agonist or temperature?

3. In fig. 4, AA-mediated currents looked to be increasing with time even in the absence of Ech A. What is the explanation for the current running-up? and have authors checked if those TREK-2 and TRAAK currents were blocked by their specific inhibitors?

4. Is the facilitation of TREK-2 and TRAAK current activation by Ech A in a concentration-dependent manner?

Author Response

1. The inhibition of TRPV3 by Ech A looked quite slow. How specific for Ech A affecting or inhibiting some other members of thermoTRPV subfamily such as TRPV1, 2 and 4? It will be interesting to know if Ech A can be used as a tool molecule for the TRPV subfamily.

Response: According to the comment, we examined the effect of Ech A on TRPV1 over expressed in HEK293 cells. Interestingly, the capsaicin-induced TRPV1 current was partly increased, not decreased, by Ech A. Such effect was different from the inhibition of TRPV3 by Ech A. The application of Ech A alone did not activate the TRPV1 current. The intriguing result is newly included in the revised manuscript Figure 2. Also the relevant discussion is included at the end of Discussion.

About the other TRPV family members, we also tried to examined the effect on TRPV4 over expressed in HEK293 cells. Unfortunately, for unknown reason, the TRPV4 current induced by the specific agonist GSK1069790A was not consistently modulated by the application of Ech A; from partial inhibition to partial activation. Since the inconsistent effects might confuse the readers, the results of TRPV4 was not included in the revised manuscript. Please understand the situation of limited time for revision.

2. Related to above, can Ech A inhibit TRPV3 current activated by other specific agonist or temperature?

Response: According to the comment, we also examined the TRPV3 current activated by another agonist, drofenine (0.5 mM). The drofenin-activated TRPV3 current was also effectively inhibited by Ech A. The description is included in the revised Result (line 126-127).

3. In fig. 4, AA-mediated currents looked to be increasing with time even in the absence of Ech A. What is the explanation for the current running-up? and have authors checked if those TREK-2 and TRAAK currents were blocked by their specific inhibitors?

Response: The impression of creeping increase before the treatment with Ech A could be due to the slow response (activation) of TREK-2/TRAAK current to AA.  Nevertheless, if the reviewer look into the revised Figure 5 (a) and (c), it can be found that the steady-state activation by AA was reached (note the downward arrows with (2) in the figure). The inhibition of TREK family K+ channels by their blockers (norfluoxetine (Nfx) or ruthenium red (RR)) could be found in the traces of Fig. 7.  Furthermore, the large amplitude of AA-actiavted outward current with the reversal potential of -85 ~ -90 mV (Fig. 5(b), (d)) would reflect the K+ selective channels in the TREK-2 or TRAAK over-expressed condition.

4. Is the facilitation of TREK-2 and TRAAK current activation by Ech A in a concentration-dependent manner?

Response: We have actually examined the effects of Ech A from submicromolar ranges to 30 uM. Unfortunately, however, more rigorous pharmacological analysis of the concentration-dependent effect to obtain the half-effective concentration (EC50) could not be conducted since the facilitating effects at the relatively low concentrations were widely variable. The description is included in the revised manuscript (Line 200-203).

Reviewer 4 Report

This MS described the effect of Echinochorme A on various ion channels using the patch clamp technique. To explain anti-inflammatory and analgesic effects by Echinochrome A, the authors applied overexpressed system of known skin channels including TRPV3, Orai1 and K2P channels. The results are convincing and interpretation are excellent. I believe this MS must be revised version and the responses are well-documented (based on red-font section). I do not see any further consideration for the publication.

Author Response

I greatly appreciate the reviewer's favorable evaluation and kind opinion.

Round 2

Reviewer 1 Report

I read the manuscript after the revision. I don't have all the answers. I have a few questions and comments.

1. Literature references [36-38] refer to data from preclinical studies. Please provide correct references for use as a medicinal product in humans (e.g. https://doi.org/10.1016/j.jep.2019.111933) and not in animals.

2. In the "Materials and Methods" section, please indicate the date of extraction/semisynthesis of echinochrome, indicating the literary reference for the extraction/semisynthesis method or batch, production date and expiration date, the manufacturer of the preparation Gistchrom. Echinochrome is an unstable quinoid pigment.

3. Include your response to question #10 (What is the toxicity of Echinnochrome A when applied topically? Provide data or literary reference.) in the text of the manuscript. Quinoid pigments are highly reactive molecules. Their risks must be assessed for application to the skin.

4. Please provide data (your or literary) that could confirm your sentence "the chemistry and pharmacokinetics of EchA after oral administration revealed that it complied with the Lipinski Rule of Five ...".

Author Response

I read the manuscript after the revision. I don't have all the answers. I have a few questions and comments.

Response: Thank you very much for the kind comments.

  1. Literature references [36-38] refer to data from preclinical studies. Please provide correct references for use as a medicinal product in humans (e.g. https://doi.org/10.1016/j.jep.2019.111933) and not in animals.             Response: There are several reports on the human application of Histochrome. The references were replaced with the human clinical studies. Also, in the revised manuscript, we now add a recent review paper, Ref.40. (Kim HK, Vasileva EA, Mishchenko NP, Fedoreyev SA, Han J. Multifaceted Clinical Effects of Echinochrome. Mar Drugs. 2021 Jul 26;19(8):412. doi: 10.3390/md19080412. PMID: 34436251; PMCID: PMC8400489.
  2.  In the "Materials and Methods" section, please indicate the date of extraction/semisynthesis of echinochrome, indicating the literary reference for the extraction/semisynthesis method or batch, production date and expiration date, the manufacturer of the preparation Histchrom. Echinochrome is an unstable quinoid pigment.                                      Response: The description about Ech A is now rewritten to include the information requested by the reviewer (revised section 4.2). In the aqueous solution, Ech A is unstable only when exposed to the air and light. It is clarified that the stock solution (Histochrome) was tightly sealed and light-protected before the experiment (revised Methods section 4.2). The process of histochrome obtaining is described in the newly added Ref 70 (Mishchenko et al. Molecules, 2020). The preparation obtained by this method stays stable in sealed ampoules for several years what was proved with investigation [Ref. 70]
  3. Include your response to question #10 (What is the toxicity of Echinnochrome A when applied topically? Provide data or literary reference.) in the text of the manuscript. Quinoid pigments are highly reactive molecules. Their risks must be assessed for application to the skin. Response: A recent study by Mishchenko et al [Ref. 70] suggests that the toxicity of Ech A and its metabolites are insignificant. It is now included in the last section of Discussion (Ref. 70).
  4. Please provide data (your or literary) that could confirm your sentence "the chemistry and pharmacokinetics of Ech A after oral administration revealed that it complied with the Lipinski Rule of Five ...".                                 Response: The sentence mentioned in our previous reply to the reviewer's comment was cited from a reference p.2, line 7-9 (Rubilar et al. https://www.ncbi.nlm.nih.gov/pmc/articles/PMC8151293/pdf/marinedrugs-19-00267.pdf ). However, since the 'Rule of Five' is not directly relevant with the current study, I do not include this in the main text.

Author Response

The authors made substantial changes in both content and presentation. The revised manuscript has improved with the additional information. I suggest the following changes

Response: Thank you very much for the kind comments and favorable opinions.

  1. Methods: For the new Section 4.2, please provide information on all ingredients in the commercial drug Histochrome® , and how it was used to make the testing solution. In particular, please comment on solubility of Histochrome® and on the final Na+ concentration in testing solutions. In Section 4.3, please add information on how the testing solutions were used—were they used to replace the extracellular solution during patch clamp recordings?

Response: For the experiment, Histochrome® was directly diluted to the bath solution (NT solution) lower than 1:1000 ratio (e.g. 24 microL of Histochrome®/30 mL of NT solution). Also, since the bath was continuously perfused with experimental extracellular NT solution, Ech A was included in the bath perfusate for the experiment. It is now described in the revised Methods section 4.2 and 4.3.

  1. Title: Since the study is on selected ion channels that might be related to skin health, the title should more accurately reflect that. One suggestion: “Multiple effects of Echinochrome A on selected ion channels implicated in skin physiology”. If it is indeed true that the authors were the “first to report pharmacological effects of Ech A on ion channels” (Section 5. Conclusions), the phrase “implicated in skin physiology” can be omitted without diminishing the impact of this study.

Response: the title was changed as kindly suggested by the reviewer. “Multiple effects of Echinochrome A on selected ion channels implicated in skin physiology”

Reviewer 3 Report

Authors have sufficiently addressed previous my concerns and questions by adding new data that appear in this revision that I am satisfied with.

Round 3

Reviewer 1 Report

Authors have not completely addressed my recommendations.

Author Response

We have revised the manuscript again, and conducted additional experiments according to the comments by other reviewers.